# Decreased cardiac pacemaking and attenuated β-adrenergic response in TRIC-A knockout mice

Manabu Murakami[1]*, Yuichi Toyama[2], Manabu Yonekura[2], Takayoshi Ohba[3], Yasushi Matsuzaki[4], Daisuke Sawamura[4], Agnieszka M. Murakami[1], Miyuki Nishi[5], Shirou Itagaki[6], Hirofumi Tomita[2], Hiroshi Takeshima[5]

1 Department of Pharmacology, Hirosaki University Graduate School of Medicine, Hirosaki, Japan,
2 Department of Cardiology and Nephrology, Hirosaki University Graduate School of Medicine, Hirosaki, Japan, 3 Department of Cellular Physiology, Akita University Graduate School of Medicine, Akita, Japan,
4 Department of Dermatology, Hirosaki University Graduate School of Medicine, Hirosaki, Japan,
5 Department of Biological Chemistry, Kyoto University Graduate School and Faculty of Pharmaceutical Sciences, Kyoto, Japan, 6 Collaboration Center for Community and Industry, Sapporo Medical University, Sapporo, Japan

* mmura0123@hotmail.co.jp

**Data Availability Statement:** All relevant data are within the manuscript and its Supporting Information files.

## Abstract

Changes in intracellular calcium levels in the sinus node modulate cardiac pacemaking (the calcium clock). Trimeric intracellular cation (TRIC) channels are counterion channels on the surface of the sarcoplasmic reticulum and compensate for calcium release from ryanodine receptors, which play a major role in calcium-induced calcium release (CICR) and the calcium clock. TRIC channels are expected to affect the calcium clock in the sinus node. However, their physiological importance in cardiac rhythm formation remains unclear. We evaluated the importance of TRIC channels on cardiac pacemaking using TRIC-A-null (TRIC-A$^{-/-}$) as well as TRIC-B$^{+/-}$ mice. Although systolic blood pressure (SBP) was not significantly different between wild-type (WT), TRIC-B$^{+/-}$, and TRIC-A$^{-/-}$ mice, heart rate (HR) was significantly lower in TRIC-A$^{-/-}$ mice than other lines. Interestingly, HR and SBP showed a positive correlation in WT and TRIC-B$^{+/-}$ mice, while no such correlation was observed in TRIC-A$^{-/-}$ mice, suggesting modification of the blood pressure regulatory system in these mice. Isoproterenol (0.3 mg/kg) increased the HR in WT mice (98.8 ± 15.1 bpm), whereas a decreased response in HR was observed in TRIC-A$^{-/-}$ mice (23.8 ± 5.8 bpm), suggesting decreased sympathetic responses in TRIC-A$^{-/-}$ mice. Electrocardiography revealed unstable R-R intervals in TRIC-A$^{-/-}$ mice. Furthermore, TRIC-A$^{-/-}$ mice sometimes showed sinus pauses, suggesting a significant role of TRIC-A channels in cardiac pacemaking. In isolated atrium contraction or action potential recording, TRIC-A$^{-/-}$ mice showed decreased response to a β-adrenergic sympathetic nerve agonist (isoproterenol, 100 nM), indicating decreased sympathetic responses. In summary, TRIC-A$^{-/-}$ mice showed decreased cardiac pacemaking in the sinus node and attenuated responses to β-adrenergic stimulation, indicating the involvement of TRIC-A channels in cardiac rhythm formation and decreased sympathetic responses.

**Funding:** This research was sponsored in part by Grants-in-Aid for Scientific Research from JSPS, KAKENHI (17K08527(MM), 17H04319(MM), 16K09489(HT), and 20K07255(MM)).

**Competing interests:** The authors have declared that no competing interests exist.

## Introduction

The cardiac pacemaker is regulated by the surface membrane potential and changes in intracellular calcium levels, which are referred to as the membrane and calcium clocks, respectively. In spontaneously firing sinus node action potentials, these two clocks work together through numerous interactions modulated by membrane voltage, subsarcolemmal calcium, and protein phosphorylation [1]. The calcium clock is mainly affected by three factors, i.e., calcium influx through voltage-dependent calcium channels and intracellular calcium release and uptake. Calcium release from internal calcium stores is mainly mediated by two major intracellular calcium channels, i.e., inositol 1,4,5-trisphosphate receptors ($IP_3Rs$) and ryanodine receptors (RyRs). RyRs are expressed in excitable cells, such as the sinus node, and play major roles in calcium-induced calcium release (CICR) from intracellular calcium stores [2, 3]. In the sinus node, calcium release through RyRs from the sarcoplasmic reticulum (SR) yields negative potentials that disturb subsequent calcium release in the lumen.

Counterion movement due to trimeric intracellular cation (TRIC) channels located on the SR membrane in the sinus node is thought to balance this negative membrane potential, resulting in cancellation of electronic disturbance due to calcium release from RyRs [4]. As TRIC channels play a significant role in maintaining ion balance in the SR, they may modulate the calcium clock by affecting calcium-related ion currents in the sinus node. Which of the two known TRIC isoforms (TRIC-A and TRIC-B) plays a significant role in calcium movement in the sinus node is not known. Furthermore, the roles of TRIC channels in the sinus node and cardiac pacemaking remain obscure. Therefore, the present study evaluated the importance of TRIC channels in the sinus node with genetically manipulated TRIC-A$^{-/-}$ and TRIC-B$^{+/-}$ mice.

## Materials and methods

Detailed methods and supplemental information are available in the online version of this article.

### Ethical approval

All experimental procedures were approved by the Institutional Animal Care and Research Advisory Committee of the Hirosaki University School of Medicine (Approval Number: 16T015).

### Model animals

Wild-type (WT), TRIC-A$^{-/-}$, and TRIC-B$^{+/-}$ mice, 12–20 weeks old, were used in all experiments. TRIC-A$^{-/-}$ and TRIC-B$^{+/-}$ mice were generated as described previously [4]. As reported previously, Tric-b-null is lethal, so haplodeficient TRIC-B$^{+/-}$ mice were used [4, 5]. To isolate hearts, mice were anesthetized with an intraperitoneal injection of a mixture of medetomidine hydrochloride (0.315mg/kg), midazolam (2.0mg/kg), and butorphanol tartrate (2.5mg/kg). To prevent any pain associated with injections, mice were first anesthetized by inhalation of 80% carbon dioxide and 20% oxygen prior to the intraperitoneal injection.

### RNA isolation and RT-PCR

The sinus node was dissected under a dissection microscope with observation of cardiac automaticity (MZ8; Leica GmbH, Wetzler, Germany). Total RNA was extracted from the sinus nodes using an RNeasy Kit (Qiagen, Valencia, CA) [6]. Reverse transcription was performed with oligo-dT primer and SuperScript IV reverse transcriptase (Invitrogen, San Diego, CA).

Comparative reverse transcription polymerase chain reaction (RT-PCR) was performed using EXTaq (Takara, Otsu, Japan) with 30–38 PCR cycles. Primers used in RT-PCR analyses are shown in S1 Table in S1 File.

## Western blotting

Western blotting analyses were performed as described previously [7]. Briefly, protein samples from the sinus nodes were homogenized and lysed in 50 mM Tris (pH 7.5), 140 mM NaCl, and 5 mM ethylenediaminetetraacetic acid (EDTA) with protease inhibitor cocktail (Complete Mini; Sigma-Aldrich, St. Louis, MO). Aliquots (100 μg) of the homogenate from each mouse were resolved by 7.5% SDS-PAGE and subjected to Western blotting. To evaluate CREB and pCREB expression, and phospholamban (PLN) and phosphorylated -phospholamban (p-PLN), we excised the sinus nodes 10 min after intraperitoneal injection of isoproterenol (0.5 mg/kg).

## Blood pressure measurement

All physiological studies were conducted between 10:00 and 16:00 at room temperature (23˚C). The mice were kept in a warmed chamber at 37˚C. Systolic blood pressure (SBP) was measured in conscious mice with the tail-cuff method (MK-1030; Muromachi Kikai Co., Ltd., Tokyo, Japan).

## Electrocardiography

The mice were anesthetized as described previously [6] during implantation of the electrical lead into the back. Electrocardiography (ECG) was recorded with a preamplifier (MEG-5200; Nihon Kohden, Tokyo, Japan) through the electrical lead, digitized with a Power Lab system, and analyzed with LabChart 8 (AD Instruments, Dunedin, New Zealand).

## Spontaneously beating atria

The right atrium was dissected free of ventricular tissue and placed in an oxygenated 37˚C tissue bath containing Tyrode's solution (123.8 mM NaCl, 5.0 mM KCl, 2.0 mM $CaCl_2$, 1.2 mM $MgCl_2$, 25.0 mM $NaHCO_3$, and 11.2 mM glucose). Isometric contractile force was measured using a force transducer (CD200; Nihon Kohden) [7].

## Cardiac action potentials in the sinus node

The spontaneously beating right atrium was mounted in the organ bath (10 mL) and perfused continuously with oxygenated Tyrode's solution at 23˚C. The sinus node region was dissected, and the spontaneously beating sinus node region was mounted in the organ bath. Conventional glass pipettes filled with 3.0 M KCl (tip resistance: 20–30 MΩ) were used to record spontaneous action potentials with an amplifier (MEZ-8301; Nihon Kohden) for analyses, as described previously [8].

## Statistical analyses

All data are shown as the mean ± standard error of the mean (SEM). The analyses were performed with JMP Pro 13 (SAS Institute, Cary, NC) using the Student's *t* test. In all analyses, $P < 0.05$ was taken to indicate statistical significance.

## Results

### RT-PCR analyses

As TRIC channels were estimated to affect calcium handling in the pacemaker cells in the sinus node, we analyzed the expression of TRIC-related molecules in the sinus node by RT-PCR analyses. RT-PCR analyses revealed slight expression of TRIC-A gene. TRIC-A-specific oligo DNA primers confirmed null mutation in the TRIC-A$^{-/-}$mice, while no significant changes in the TRIC-A gene were detected in the TRIC-B$^{+/-}$SA node (Fig 1A). We also confirmed expression of the TRIC-B gene in the wild-type SA node, as well as in the TRIC-A-deficient SA node. As expected, the TRIC-B $^{+/-}$SA node showed decreased TRIC-B gene expression. After ablation of the TRIC-A gene, there were no significant changes in TRIC-B expression, and TRIC-B haplodeficiency resulted in no significant changes in TRIC-A expression, suggesting that expression of each TRIC channel is regulated independently.

The sinus nodes in the TRIC-A$^{-/-}$mice showed slightly increased β1-adrenergic receptor expression compared to the WT controls ($1.60 \pm 0.07$ vs. $2.07 \pm 0.14$, $n = 4$, respectively; $P < 0.05$), while no significant increase was observed in the TRIC-B$^{+/-}$SA nodes. As the β1-adrenergic receptor is related to sympathetic nerve regulation, these results suggest the involvement of TRIC-A channels in sympathetic nerve regulation. We also examined the expression of adrenergic β$_2$-receptor, and no significant changes were observed. No significant differences were noted in the expression of parasympathetic M2 receptor or sodium-calcium exchanger-1 (NCX1) in the sinus nodes of TRIC-A$^{-/-}$or TRIC-B $^{+/-}$mice. Next, we examined the expression profiles of RyRs and voltage-dependent calcium channels. No expression of RyR1 was observed, while both PyR2 and RyR3 expression were detected in the SA nodes. The L-type voltage-dependent calcium channels, CaV1.2 and CaV1.3, were confirmed to be expressed.

Next, we evaluated the expression of KCa1.1 because large-conductance Ca$^{2+}$-activated K$^+$ (BK) channels encoded by KCa1.1 may be stimulated by changes in intracellular calcium level due to TRIC gene manipulations. The sinus nodes of the TRIC-A$^{-/-}$mice showed increased

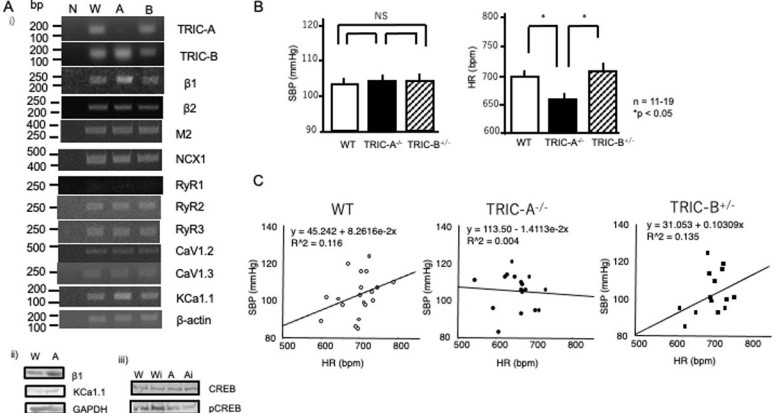

**Fig 1. A. RT-PCR and Western blotting.** i) RT-PCR. RT-PCR analyses of the sinus nodes of WT (W), TRIC-A$^{-/-}$(A), and TRIC-B$^{+/-}$(B) mice (A). $n = 4$. N, negative control. Primer sets are indicated. ii). Western blotting analyses. Western blotting analyses of adrenergic β1 receptor (β1), KCa1.1, and GAPDH expression in the sinus nodes of WT (W) and TRIC-A$^{-/-}$mice (A). CREB and pCREB were evaluated before and after injection of isoproterenol. iii) Isoproterenol-induced phosphorylation of CREB in wild-type and TRIC-A$^{-/-}$SA nodes. WT, wild-type; Wi, WT mice given isoproterenol; Ai, TRIC-A$^{-/-}$mice given isoproterenol. **B. Systolic blood pressure and heart rate** Basal systolic blood pressure (SBP) and heart rate (HR) in wild-type (WT) (open bars), TRIC-A$^{-/-}$(solid bars), and TRIC-B$^{+/-}$mice (hatched bars). $n = 11-19$. $P < 0.05$. **C. Scatterplot of systolic blood pressure and heart rate**. SBP and HR data of wild-type (WT) (circles, left panel), TRIC-A$^{-/-}$(dots, middle panel), and TRIC-B$^{+/-}$mice (solid squares, right panel) were plotted. Calculated regression lines are indicated.

expression of KCa1.1 compared to the WT mice (1.20 ± 0.21 vs. 1.78 ± 0.30, $n = 4$, respectively; $P < 0.05$), while TRIC-B$^{+/-}$ mice showed no significant changes. We further explored the expression of hyperpolarization-activated cyclic nucleotide-gated (HCN) 2 and 4 channels, sarco/endoplasmic reticulum $Ca^{2+}$-ATPase (SERCA), inositol trisphosphate receptor-type2 (IP3R2), stromal interaction molecule (STIM)1, Orai1, Orai2 and phospholamban (PLN) in the SA nodes.

Overall, RT-PCR analyses revealed slightly increased β1-adrenergic receptor expression in the sinus nodes of TRIC-A$^{-/-}$ mice, suggesting that gene manipulation of TRIC-A channels may affect sympathetic nerve regulation.

## Western blotting analyses

As RT-PCR analyses revealed significant changes only in TRIC-A$^{-/-}$ mice, we focused on the differences between TRIC-A$^{-/-}$ and WT mice. Western blotting analyses revealed an increase in β1-adrenergic receptor protein expression in the TRIC-A$^{-/-}$ sinus nodes compared to those of WT controls (1.08± 0.04 vs. 1.65 ± 0.10, $n = 6$, respectively; $P < 0.05$) (Fig 1 and S1 Fig in S1 File). The sinus nodes of TRIC-A$^{-/-}$ mice also showed increased KCa1.1 protein expression compared to those of WT mice (0.92 ± 0.04 vs. 0.71 ± 0.05, $n = 6$, respectively; $P < 0.05$) (Fig 1 and S1 Fig in S1 File). GAPDH was used as an internal control. These results suggest the involvement of TRIC channels in sympathetic nerve regulation. Therefore, we examined increases in pCREB in the sinus node to evaluate the β-adrenergic response. Isoproterenol injection resulted in a significant increase in phosphorylated-CREB level in the WT SA nodes, while a marginal response was observed in TRIC-A$^{-/-}$ SA nodes compared to WT (1.38 ± 0.08 vs. 1.03 ± 0.07-fold, $n = 6$, respectively; $P < 0.05$) (Fig 1 and S1 Fig in S1 File). Total CREB levels were also evaluated (CREB), and the results suggested a decreased β1-adrenergic response in TRIC-A$^{-/-}$ SA nodes. We also examined phosphorylation of phospholamban (S1 Fig in S1 File). Isoproterenol injection resulted in a significant increase in phosphorylated-phospholamban level in the WT SA nodes, while a limited response was observed in TRIC-A$^{-/-}$ SA nodes compared to WT (2.23 ± 0.06 vs. 1.17 ± 0.04-fold, $n = 6$, respectively; $P < 0.05$). The increased β1-adrenergic receptor expression may be a compensation for decreased PKA-dependent signal transduction.

## Systolic blood pressure and heart rate

We found no significant differences in basal blood pressure recordings between WT and TRIC-A$^{-/-}$ mice (Fig 1). Yamazaki *et al.* reported increased SBP in TRIC-A$^{-/-}$ mice due to increased vascular resistance [5]. However, because this increase was relatively limited, we re-analyzed the relationship between SBP and HR in TRIC-A$^{-/-}$ mice. Conversely, the basal HR in the TRIC-A$^{-/-}$ mice (660.3 ± 9.6 bpm, $n = 15$; $P < 0.05$) was significantly lower than that in WT mice (698.5 ± 10.5 bpm, $n = 19$), in line with that previous study (Fig 1). These discrepancies compared to the previous study suggest that the decrease in HR associated with TRIC-A gene ablation may have been due to other unknown factors rather than increased SBP.

As there is generally a positive correlation between SBP and HR [9, 10], we plotted SBP and HR in the WT and TRIC-A$^{-/-}$ mice (Fig 1). Both WT and TRIC-B$^{+/-}$ mutant mice showed positive correlations between SBP and HR (Fig 1), while no such correlation was observed in TRIC-A$^{-/-}$ mice (Fig 1), indicating modification of the blood pressure regulatory system in TRIC-A$^{-/-}$ mice. This lack of correlation was difficult to explain based merely on high resistance in the peripheral vessels, as previously reported [5].

## Pharmacological manipulations

**Effects of isoproterenol, caffeine, and dantrolene.** Next, we examined pharmacological manipulations with a β-adrenergic agonist, as the sympathetic nervous system is a major autonomic nervous system, which regulates the cardiovascular system, including blood pressure, heart rate, and vascular resistance. Isoproterenol (a nonselective β-agonist) was used as a sympathetic nerve stimulus (Fig 2). The intraperitoneal administration of isoproterenol (0.3 mg/kg) resulted in decreased SBP in WT (–32.8 ± 6.2 mmHg, $n$ = 9), TRIC-A$^{-/-}$ (–26.8 ± 6.3 mmHg, $n$ = 10), and TRIC-B$^{+/-}$ mice (–22.6 ± 6.3 mmHg, $n$ = 10). There were no significant differences among these three groups. Notably, isoproterenol increased the HR in WT mice (98.8 ± 15.1 bpm, $n$ = 9), whereas it induced only a slight increase in TRIC-A$^{-/-}$ mice (23.8 ± 5.8 bpm, $n$ = 10, $P$ < 0.05), while TRIC-B$^{+/-}$ mice showed increased HR (64.0 ± 15.0 bpm, $n$ = 10), indicating that TRIC-A$^{-/-}$ mice have decreased responsiveness to β-agonists, which strongly suggests modification of the sympathetic regulation system.

As only TRIC-A$^{-/-}$ mice showed a significant decrease in HR and a significant decrease in HR changes in response to isoproterenol, we focused on the difference between WT and TRIC-A$^{-/-}$ mice. We examined the effects of pharmacological manipulation with caffeine (calcium release modulator from RyR) and dantrolene (a blocker of RyR) in WT and TRIC-A$^{-/-}$ mice. The administration of a single dose of caffeine typically increases blood pressure (3–14 mmHg in SBP) in humans [11, 12]. In the present study, the intraperitoneal administration of caffeine (10 mg/kg) slightly altered the SBP in WT (4.2 ± 2.4 mmHg, $n$ = 6) and TRIC-A$^{-/-}$ mice (9.5 ± 3.8 mmHg, $n$ = 6) (Fig 2). There were no significant differences between the two groups. Caffeine increased HR in the WT (52.8 ± 38.1 bpm, $n$ = 6) and TRIC-A$^{-/-}$ mice (55.8 ± 21.8 bpm, $n$ = 6), and the difference between the two groups was not significant (Fig 2). Dantrolene depresses excitation–contraction coupling in the

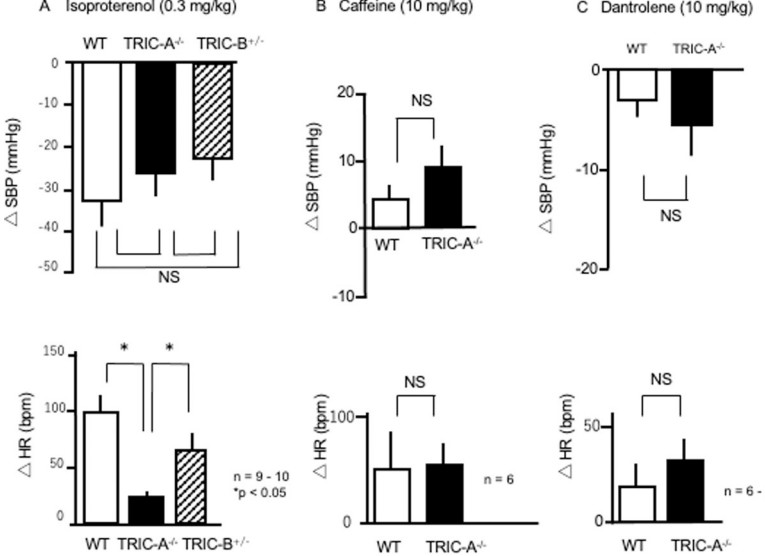

**Fig 2. Changes in systolic blood pressure and heart rate due to pharmacological manipulation.** Systolic blood pressure (SBP) and heart rate (HR) after administration of isoproterenol (0.3 mg/kg; A), caffeine (10 mg/kg; B), and dantrolene (10 mg/kg; C) in wild-type (WT) (open bar), TRIC-A$^{-/-}$ (solid bar), and TRIC-B$^{+/-}$ (hatched bar) mice. **A.** Isoproterenol significantly decreased SBP in all three groups (upper panel). WT and TRIC-B$^{+/-}$ mice showed significant increases in HR in response to isoproterenol. TRIC-A$^{-/-}$ mice showed a marginal increase in HR due to isoproterenol. $n$ = 9–10. $P$ < 0.05. **B.** Caffeine induced a slight increase in SBP in WT mice and TRIC-A$^{-/-}$ mice. $n$ = 6 in each group. **C.** Dantrolene slightly decreased SBP in WT mice and TRIC-A$^{-/-}$ mice. Values are the mean ± standard error of the mean. $n$ = 6–7. SBP, systolic blood pressure; HR, heart rate.

skeletal muscle by antagonizing RyRs, resulting in inhibition of calcium release from SR. Intraperitoneal administration of dantrolene (0.3 mg/kg) resulted in slight decreases in SBP in WT (–2.7 ± 1.8 mmHg, $n$ = 6) and TRIC-A$^{-/-}$ mice (–5.3 ± 3.3 mmHg, $n$ = 7) and increases in HR in WT (21.4 ± 11.7 bpm, $n$ = 6) and TRIC-A$^{-/-}$ mice (34.8 ± 9.8 bpm, $n$ = 7) (Fig 2). We also examined the effects of α1 agonist. TRIC-A$^{-/-}$ mice showed decreased responsiveness, which may have been due to the previously reported increase in vascular resistance. We also examined the effects of calcium antagonists, which are widely used as antihypertensive and antiarrhythmic class-III drugs. TRIC-A$^{-/-}$ mice showed decreased responsiveness, in accordance with the previously reported increase in vascular resistance.

**Electrocardiography.** In a rigorous evaluation of cardiac rhythm formation, ECG, which accurately determines electrical RR interval, is more accurate than blood pressure analyses, which depends on changes in pressure. Fig 3 shows representative ECG traces of WT, TRIC-A$^{-/-}$, and TRIC-B$^{+/-}$ mice. During the basal recording (ca. 5 min), TRIC-A$^{-/-}$ mice showed elongated RR interval compared to WT or TRIC-B$^{+/-}$ mice, resulting in decreased HR (708.9 ± 11.7, 693.8 ± 13.4 bpm, 638.2 ± 20.9, $n$ = 9–11, respectively; $P$ < 0.05) (Fig 3), as expected based on the results of blood pressure analyses. Notably, 42.9% of TRIC-A$^{-/-}$ mice (12 of 28 recordings) showed unstable RR interval and P wave dropout, indicating a sinus pause with normal ECG waves, suggesting supraventricular bradycardia. Misfiring P waves and subsequent QRS waves were suddenly observed. Some showed atrioventricular block.

**Modified heart rate variability in TRIC-A$^{-/-}$ mice.** Collected ECG waves of the three lines of mice were further analyzed with heart rate variability (HRV) software. No significant differences were observed in any of the ECG parameters, including PQ and QT intervals or P and QRS durations, among the three lines. Fig 3 shows representative Poincaré plots for WT, TRIC-A$^{-/-}$, and TRIC-B$^{+/-}$ mice. WT and TRIC-B$^{+/-}$ mice exhibited regular RR intervals, whereas the TRIC-A$^{-/-}$ mice had quite unstable RR intervals. Power spectral analyses did not show any significant differences among the three groups. For further HRV-related frequency analyses, we focused on TRIC-A$^{-/-}$ mice. We found no significant differences in high frequency

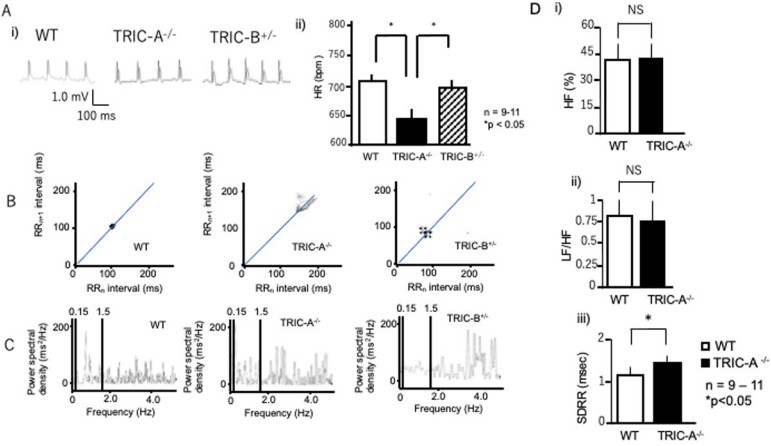

**Fig 3. Electrocardiography and heart rate variability analyses. A.** i) Representative electrocardiography (ECG) traces of WT, TRIC-A$^{-/-}$, and TRIC-B$^{+/-}$ mice. ii) Statistical analyses of calculated HR in WT (open bar), TRIC-A$^{-/-}$ (solid bar), and TRIC-B$^{+/-}$ (hatched bar) mice. $n$ = 9–11. $P$ < 0.05. **B.** Representative Poincaré plots of RR ($n$) on the x-axis vs. RR ($n$+1) on the y-axis of WT, TRIC-A$^{-/-}$, and TRIC-B$^{+/-}$ mice. The TRIC-A$^{-/-}$ mice had relatively long and unstable RR intervals. **C.** Power spectra by frequency domain analyses of WT, TRIC-A$^{-/-}$, and TRIC-B$^{+/-}$ mice. **D.** Heart rate variability (HRV) parameters, HF (i), LF/HF (ii), and SDRR (iii) in WT (open bars) and TRIC-A$^{-/-}$ mice (solid bars). There were no significant differences in HF or LH/HF between WT and TRIC-A$^{-/-}$ mice. TRIC-A$^{-/-}$ mice had a longer SDRR than WT mice. $n$ = 9–11. $P$ < 0.05.

power (HF) or low frequency power/high frequency power (LF/HF) ratio between the WT and TRIC-A$^{-/-}$ mice. However, the basal standard deviation of normal RR intervals (SDRR) was larger in the TRIC-A$^{-/-}$ mice (1.46 ± 0.11 ms, $n$ = 9, $P < 0.05$) than in the WT mice (1.17 ± 0.07 ms, $n$ = 11). As SDRR is calculated as the standard deviation of the RR interval, the increase in SDRR was consistent with the unstable RR observed in the Poincaré plots of the TRIC-A$^{-/-}$ mice. Taken together, the results of ECG analyses revealed unstable pacemaking in TRIC-A$^{-/-}$ mice, suggesting an important role of TRIC-A in the cardiac clock system.

**Atrial contraction.** As modified sympathetic signal transduction in the TRIC-A$^{-/-}$ heart may cause decreased responsiveness to β-adrenergic stimulation, we assessed sympathetic responsiveness by measuring inotropic responses to the β-adrenergic agonist, isoproterenol, in isolated spontaneously beating right atria. Fig 4 shows typical force traces under basal conditions and in response to isoproterenol (100 nM). There are dose-dependent changes in the response to isoproterenol (10–100 nM) in WT atria and TRIC-A$^{-/-}$ atria. WT atria showed increased beat rate and force in response to isoproterenol, while only limited increases were seen in HR and force in TRIC-A$^{-/-}$ atria. Statistical analyses showed dose-dependent changes in force and HR in the WT atria, whereas both force and HR exhibited limited responsiveness in the TRIC-A$^{-/-}$ atria, suggesting that the TRIC-A$^{-/-}$ atria have decreased responsiveness to β-adrenergic stimulation.

**Action potential recordings.** To examine the importance of TRIC-A channels in cardiac pacemaking, it is straightforward to analyze action potentials at the sinus node. Fig 5 shows typical action potential traces for WT and TRIC-A$^{-/-}$ sinus nodes. The action potential in the sinus node showed slow depolarization changes in both groups. TRIC-A$^{-/-}$ hearts showed significantly decreased sinus rates compared to WT (98.8 ± 3.3 bpm vs. 123.9 ± 8.1, $n$ = 5 and 6, respectively; $P < 0.05$). Adrenergic stimulation with isoproterenol (100 nM) significantly increased the action potential formation rate in the WT sinus node. Conversely, the TRIC-A$^{-/-}$ sinus node showed a limited response to isoproterenol compared to WT (35.3 ± 8.9 bpm and 70.8 ± 10.5, $n$ = 5 and 6, respectively; $P < 0.05$). These results indicate that TRIC-A$^{-/-}$ sinus nodes had lower sensitivity to sympathetic β1-adrenergic stimulation.

Previous studies have established the importance of big-conductance potassium (BK) channels in pace-making. Paxilline (PAX), a BK antagonist, induces bradycardia in rats and mice [12–14]. The intraperitoneal administration of PAX causes significant bradycardia in mice [14]. Yamazaki *et al.* reported decreased BK channel blocker sensitivity in TRIC-A$^{-/-}$ smooth muscle, indicating the importance of TRIC-A channels in vascular tonus, probably through maintaining intracellular calcium homeostasis [5]. As KCa1.1, which encodes BK channels, is expressed in the sinus node and TRIC-A$^{-/-}$ SA node, showed increased KCa1.1 expression in

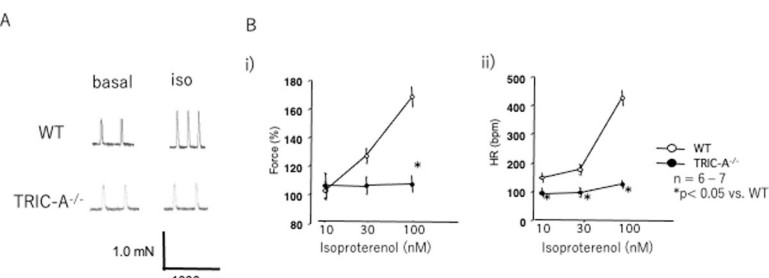

**Fig 4. Atrial contraction. A.** Representative atrial contraction of WT (upper panels) and TRIC-A$^{-/-}$ mice (lower panels). Isoproterenol (100 nM) increased heart rate and force in the WT atrium, while TRIC-A$^{-/-}$ atria showed only a limited response. **B.** Dose-dependent changes in contractility (i) and HR (ii) in response to isoproterenol (10–100 nM) of WT and TRIC-A$^{-/-}$ atria. Values are the mean ± SEM ($n$ = 6–7). $P < 0.05$, between WT atria and TRIC-A$^{-/-}$ atria. HR, heart rate; WT, wild-type.

RT-PCR and Western blotting analyses (Fig 1), we evaluated PAX sensitivity. PAX (1 μM) significantly decreased the rate of cardiac action potential formation in the hearts of WT mice, whereas only a limited effect was observed in TRIC-A$^{-/-}$ mice (–15.1% ± 4.4% and –4.4% ± 2.1%, $n = 6$ and 7, respectively; $P < 0.05$) (Fig 5), suggesting decreased sensitivity to BK channel inhibitors in TRIC-A$^{-/-}$ SA nodes. Overall, our results suggest that TRIC-A gene ablation results in modified BK channel regulation, which may be related to decreased RyR-mediated intracellular calcium homeostasis, although further analyses are required to examine this issue.

## Discussion

We analyzed the importance of TRIC channels in cardiac pacemaking. TRIC-A$^{-/-}$ mice exhibited a significantly lower HR compared with the WT and TRIC-B$^{+/-}$ mice, suggesting that TRIC-A channels are related to bradycardia. Furthermore, TRIC-A$^{-/-}$ mice showed no correlation between SBP and HR, whereas a positive correlation between SBP and HR was observed in WT and TRIC-B$^{+/-}$ mice. The TRIC-A$^{-/-}$ mice also showed limited HR changes in response to sympathetic β1-adrenergic receptor stimulation. In ECG analyses, the TRIC-A$^{-/-}$ mice showed long and unstable RR intervals and larger SDRR compared with WT mice. Isolated TRIC-A$^{-/-}$ hearts showed decreased responsiveness to isoproterenol. Action potential recording in the TRIC-A$^{-/-}$ sinus node showed a decreased sinus rate and decreased responsiveness to isoproterenol and paxilline (BK channel antagonist). Overall, our results strongly suggest the involvement of TRIC-A channels in cardiac pacemaking.

In the present study, TRIC-A$^{-/-}$ mice showed bradycardia, and their SBP was not significantly higher than that of WT mice. Yamazaki *et al.* reported increased SBP and decreased HR in young adult TRIC-A$^{-/-}$ mice (8–12 weeks old) [5]. We used relatively mature mice (12–20 weeks old), which may have affected the results. Yamazaki *et al.* concluded that the decreased HR in TRIC-A$^{-/-}$ mice was due to the baroreflex of their hypertensive condition via high vascular resistance. However, the increase in blood pressure in TRIC-A$^{-/-}$ mice was relatively small compared with the significant decrease in HR [5]. As isolated TRIC-A$^{-/-}$ atria showed a significantly lower HR than that of WT atria (Fig 4), TRIC-A gene knockout appears to directly affect action potential formation, resulting in decreased cardiac pacemaking.

Recently, Lai *et al.* reported that decreased K$^+$ currents through BK channels in the SA node result in lower HR [14]. Inhibition of BK currents by PAX, a membrane-permeable BK

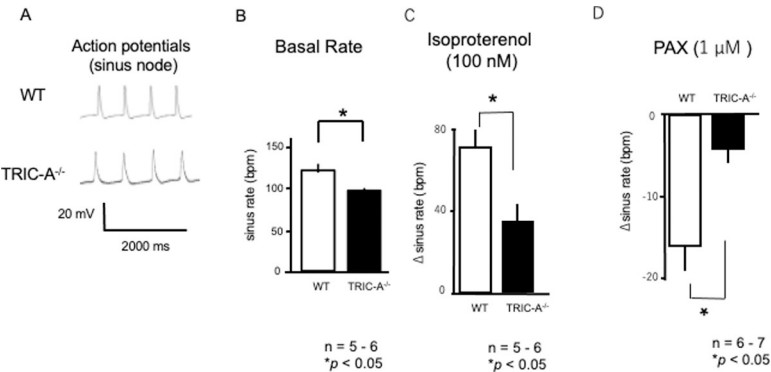

**Fig 5. Sinus nodal action potential (spontaneous sinus rate). A.** Representative action potentials of the sinus nodes of WT and TRIC-A$^{-/-}$ mice. **B.** Spontaneous sinus rates in WT (open bars) and TRIC-A$^{-/-}$ mice (solid bars). TRIC-A$^{-/-}$ SA nodes showed significantly decreased sinus rate. $n = 6$–8. $P < 0.05$. **C.** HR changes in response to isoproterenol (100 nM). TRIC-A$^{-/-}$ SA nodes showed a decreased response. $n = 5$–6. $P < 0.05$. **D.** HR changes in response to the BK antagonist, paxilline (1 μM). TRIC-A$^{-/-}$ SA nodes showed decreased response. $n = 6$–7. $P < 0.0$.

channel antagonist, slowed cardiac pacing due to elongation of the sinus interval. Furthermore, baseline firing rates from BK channel-null mice in SA nodal cells were 33% lower compared with WT SA nodal cells [15]. Previously, Yamazaki *et al.* reported deceased BK currents in the vascular smooth muscle of TRIC-A$^{-/-}$ mice [5]. Reduced counterion compensation in the SR by TRIC-A gene targeting resulted in decreased calcium-induced calcium release (CICR), which seemed to result in decreased potassium currents through BK channels in TRIC-A$^{-/-}$ vascular smooth muscle. In the present study, we also found decreased responsiveness to PAX in the TRIC-A$^{-/-}$ SA node (Fig 5). Therefore, TRIC-A gene knockout probably decreased CICR in the SA node, resulting in decreased activation of calcium-sensitive BK channels. Taken together, our data suggest that decreased basal HR in TRIC-A$^{-/-}$ mice may be related to decreased BK channel activity in the sinus node.

In the present study, the TRIC-A$^{-/-}$ SA node showed decreased responsiveness to isoproterenol. El-Ajouz *et al.* reported decreased PKA-dependent activation of RyR1 in TRIC-A$^{-/-}$ skeletal muscle [16]. The present results were consistent with their study. Recent studies showed that β-adrenergic stimulation requires a link between PKA and SR calcium cycling within the coupled-clock system [17–19]. The TRIC-A$^{-/-}$ SA node appears to have decreased CICR from RyR2 or RyR3. Therefore, decreased CICR in the TRIC-A$^{-/-}$ SA node apparently affected calcium-dependent mechanisms.

Shrestha *et al.* reported that TRIC-A interacts with the STIM1/Orai1 complex and negatively regulates STIM1/Orai1 function [20]. They reported that TRIC-A modulates SOCE and oscillatory calcium signals by affecting the STIM1/Orai1 complex. The basal cAMP level is also linked to spontaneous beating in cardiac pacemaker tissue [21]. The rabbit SA node has a higher basal level of cAMP than that of ventricular myocytes. The activities of some adenylate cyclases (ACs), such as AC1 and AC8, are also calcium dependent. Therefore, decreased calcium signals in the TRIC-A$^{-/-}$ SA node may decrease the basal cAMP level and affect the PKA cascade, resulting in a decreased β-adrenergic response. While this study was in preparation, Zhou *et al.* reported isoproterenol-induced arrhythmia in TRIC-A$^{-/-}$ mice [22]. In their study, they showed a marginal effect of isoproterenol on heart rate, which was partly consistent with the results of the present study. In addition, they reported that chronic treatment with isoproterenol resulted in cardiac fibrosis in TRIC-A$^{-/-}$ mice, which was reminiscent of chronic heart failure. It is tempting to speculate that TRIC-A may directly affect the cAMP–PKA cascade, although further analysis of the relationships of TRIC-A with cAMP and PKA is required.

The increases in sinus pause and A-V block in TRIC-A$^{-/-}$ mice were interesting because TRIC-A may be a therapeutic target. As TRIC-A gene null mutation may cause modified counterion balance in the SR, altered calcium overload in the SR in cardiomyocytes may contribute to the development of cardiac arrhythmia. In RT-PCR analyses, we detected RyR2 and RyR3 expression in the SA node (Fig 1). Thus, TRIC channels and RyRs may be potential therapeutic targets in cardiovascular diseases.

In conclusion, TRIC-A$^{-/-}$ mice showed decreased pacemaking, heart rate, and responsiveness to β-adrenergic stimulation in ECG studies. Isolated TRIC-A$^{-/-}$ atria also showed decreased cardiac action potential formation, resulting in a decreased sinus rate, as well as responsiveness to isoproterenol. Our results strongly suggest the involvement of TRIC-A channels in cardiac pacemaking and sympathetic responses.

## Supporting information

**S1 File.**
(PDF)

**S1 Text.**
(DOCX)

# Acknowledgments

We thank Ryouichi Kikuchi and Kentarou Igari for their technical assistance.
All authors contributed equally.

# Author Contributions

**Conceptualization:** Manabu Murakami.

**Funding acquisition:** Manabu Murakami, Shirou Itagaki.

**Investigation:** Manabu Murakami, Yuichi Toyama, Manabu Yonekura, Takayoshi Ohba, Agnieszka M. Murakami, Miyuki Nishi, Shirou Itagaki, Hirofumi Tomita, Hiroshi Takeshima.

**Project administration:** Manabu Murakami.

**Resources:** Yasushi Matsuzaki, Daisuke Sawamura.

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
