## [Decision Letter · Decision Letter 0]

29 Oct 2020

PONE-D-20-27381

Decreased cardiac pacemaking and attenuated β-adrenergic response in Tric-a knockout mice

PLOS ONE

Dear Dr. Murakami,

Thank you for submitting your manuscript to PLOS ONE. After careful consideration, we feel that it has merit but does not fully meet PLOS ONE’s publication criteria as it currently stands. Therefore, we invite you to submit a revised version of the manuscript that addresses the points raised during the review process.

There are some technical issues that need to be addressed that are indicated below.

We look forward to receiving your revised manuscript.

Kind regards,

Agustín Guerrero-Hernandez

Academic Editor

PLOS ONE

Journal Requirements:

2. To comply with PLOS ONE submissions requirements, in your Methods section, please provide additional information on the animal research and ensure you have included details on (1) methods of sacrifice, (2) methods of anesthesia and/or analgesia, and (3) efforts to alleviate suffering.

Reviewers' comments:

Reviewer's Responses to Questions

**Comments to the Author**

1. Is the manuscript technically sound, and do the data support the conclusions?

Reviewer #1: Yes

Reviewer #2: Partly

2. Has the statistical analysis been performed appropriately and rigorously? 

Reviewer #1: Yes

Reviewer #2: No

3. Have the authors made all data underlying the findings in their manuscript fully available?

Reviewer #1: Yes

Reviewer #2: Yes

4. Is the manuscript presented in an intelligible fashion and written in standard English?

Reviewer #1: Yes

Reviewer #2: Yes

5. Review Comments to the Author

Reviewer #1: This is a nice work by Dr. Murakami and colleagues. The authors presented solid evidence demonstrating that Tric-a, not Tric-b contributes to sinus atrial node pacemaking activity. Although the techniques used for all assays are routine, their data clearly support their conclusion that TRIC-a channel but not Tric-b is involved in cardiac rhythm formation and sympathetic nerve regulation of heart rate. Their data are of high quality. I have no any concerns or critiques.

Reviewer #2: General comments:

Murakami et al. provide evidence for a crucial role of TRICA in cardiac pace making. Using murine genetic knock-out models, the authors demonstrate significantly reduced heart rate as a result of TRICA ablation. In vivo measurements as well as experiments in isolated cardiac tissue including electrophysiological recordings in cells from the SA region demonstrate that physiologic pace making as well as adrenergic control of cardiac functions is strictly dependent on TRICA expression. The study is carefully planned and the open questions about a role of TRICA in physiologic control over heart rate are addressed by adequate and complementary methods. However, I have several issues regarding data presentation and analysis. With respect to the conclusion, I feel that the still enigmatic mechanism underlying the cardiac function of TRICA needs more detailed discussion, and some of the results, especially the measurements of PKA-mediated phosphorylation deserve more attention.

Major points:

1. The authors show convincingly that in Tric-a -/- mice not only heart rate is lower and barely increases in response to beta-adrenergic stimulation, as compared to WT or Tric-b +/-, but also beta adrenergic inotropism is largely eliminated. This is an intriguing finding and rather unexpected. The original hypothesis of the authors is that TRICA is part of the cardiac ER/SR Ca2+ handling machinery and as such part of the coupled clock in pacemaker cells. Although this is still plausible and may not be ignored, adrenergic signaling is obviously impaired at the level of PKA-dependent phosphorylation as demonstrated in Figure 1A for CREB. Since PKA-mediated phosphorylation is an essential mechanism of adrenergic control over the coupled clock machinery (see Vinogradova and Lakatta 2009; DOI: 10.1016/j.yjmcc.2009.06.014) the question arises: Can the authors exclude that the main role of TRICA is to enable proper PKA-dependent regulatory phosphorylation in cardiac cells? It is in any case very important to further strengthen the finding that genetic ablation of TRICA impairs PKA regulatory phosphorylation. Please include a more informative/ representative immunoblotting illustration together with a decent statistics on CREB phosphorylation. I strongly suggest to further corroborate this important finding also for other targets (e..g. phospholamban), and include a more in depth discussion as to how TRICA might interfere with PKA signaling. In this respect one need to consider that TRIC has recently been identified as an interaction partner of STIM1 to organize STIM/Orai complexes (Shrestha et al 2020/ DOI: 10.1371/journal.pbio.3000700), which in turn are coupled to local cAMP, PKA/Ca2+ signaling (Zhang et al 2019 /doi.org/10.1038/s41467-019-09593-0). Thus the authors may discuss the role of TRICA in SA cells beyond the original countercurrent concept or direct modulatory effects on RyR signaling.

2. Along the same lines: STIM1 is a likely interaction partner of TRICA (Shrestha et al 2020). Moreover, STIM1/Orai signaling appears crucial (Zhang et al. 2015/ doi:10.1073/pnas.1503847112). Therefore, inspection of STIM1 and also Orai expression at the RNA level is recommended and the discussion should be extended towards a potential interference with the STIM/Orai pathway.

3. The authors repetitively claim that „sympathetic nerve regulation“ is modified/impaired. From my understanding, there is no solid evidence for a role of TRICA in sympathetic neurons? By contrast, the authors demonstrate in Figure 4 almost complete elimination of isoproterenol-induced positive inotropism in isloated atria. These impressive effects can hardly be explained by sympathetic nerve regulation. The authors need to explicitly discuss the evidence for a neuronal vs a muscular/cardiac mechanism and omit all misleading statements.

Minor points:

1. The concentration of isproterenol (100 nM) should be stated also in the abstract.

2. Statistical analysis needs a more detailed description. What test(s) was/were used for comparison of multiple groups? The numbers of observations need to be stated consistently throughout the manuscript (eg. n is given with 9/10 in the text but stated as 6-15 in Figure 2A).

3. Specifically, statistics for immunoblotting data need extension. An N around 4 is in general insufficient for a decent statistical analysis. Statistical information, eg. as bar graphs, should be given in the respective figures.

4. The designation of genetic models in text an figures is highly inconsistent – please use uniform terminology. Eg. In Figure 3, four(!) different terms are used for TRICA knock out mice: TricA-null, Tric-A-null, TricA KO and Tric-a- -/- .

5. In Figure 4, representative traces of contractile force are apparently of similar absolute amplitude and share the same scaling. However, the signal to noise ratio is substantially lower in the TRICA KO atria. Do the authors have any plausible explanation for this?

6. As above: Why are the Tric-a -/- recordings in Figure 5 so noisy compared to WT?

7. The discussion may deserve remodeling – not only by including aspects mentioned above but also shortening other parts.

8. Supplementary data are not reasonably mentioned in the text. The authors should clearly indicate, mention the results illustrated in the supplement.

6. PLOS authors have the option to publish the peer review history of their article (what does this mean?). If published, this will include your full peer review and any attached files.

Reviewer #1: No

Reviewer #2: No

---

## [Author Response · Author response to Decision Letter 0]

17 Nov 2020

Our point-by-point responses to each of the comments are presented below. We thank you for your consideration of our manuscript and look forward to hearing your decision.

Sincerely,

Manabu Murakami

Department of Pharmacology, 

Hirosaki University, Graduate School of Medicine

E-mail: mmura0123@hotmail.co.jp

>>>>>>>

PONE-D-20-27381

Decreased cardiac pacemaking and attenuated β-adrenergic response in TRIC-A knockout mice

PLOS ONE

Dear Dr. Murakami,

Thank you for submitting your manuscript to PLOS ONE. After careful consideration, we feel that it has merit but does not fully meet PLOS ONE’s publication criteria as it currently stands. Therefore, we invite you to submit a revised version of the manuscript that addresses the points raised during the review process.

There are some technical issues that need to be addressed that are indicated below.

We look forward to receiving your revised manuscript.

Kind regards,

Agustín Guerrero-Hernandez

Academic Editor

PLOS ONE

Journal Requirements:

 >>>>

We have ensured that that the revised manuscript conforms to the style requirements of PLoS ONE.

2. To comply with PLOS ONE submissions requirements, in your Methods section, please provide additional information on the animal research and ensure you have included details on (1) methods of sacrifice, (2) methods of anesthesia and/or analgesia, and (3) efforts to alleviate suffering.

>>>>

Additional information has been provided regarding compliance with the regulations of the animal experiments.

 >>>>>>>

As we did not have uncropped images of some RT-PCR gels, we repeated the RT-PCR experiments. The original uncropped and unadjusted images of the RT-PCR gels and Western blots are presented in our Supporting Information Figure 5F. The Western blots in Figure 1A (for β1, KCa1.1, and GAPDH) were cut from original transferred membranes and subjected to immunoreaction because of the relatively low dilution of antibodies (200–500-fold). Therefore, they are original blots, whereas they are small.

>>>>>

The manuscript has been revised in accordance with the Supporting Information guidelines.

Reviewers' comments:

Reviewer's Responses to Questions

Comments to the Author

1. Is the manuscript technically sound, and do the data support the conclusions?

Reviewer #1: Yes

Reviewer #2: Partly

>>>>>>

The manuscript has been revised accordingly.

2. Has the statistical analysis been performed appropriately and rigorously?

Reviewer #1: Yes

Reviewer #2: No

>>>>>>>>

The performance of the statistical analysis has been addressed in the revised manuscript according to the reviewer’s comment.

3. Have the authors made all data underlying the findings in their manuscript fully available?

Reviewer #1: Yes

Reviewer #2: Yes

4. Is the manuscript presented in an intelligible fashion and written in standard English?

Reviewer #1: Yes

Reviewer #2: Yes

5. Review Comments to the Author

Reviewer #1: This is a nice work by Dr. Murakami and colleagues. The authors presented solid evidence demonstrating that TRIC-A, not Tric-b contributes to sinus atrial node pacemaking activity. Although the techniques used for all assays are routine, their data clearly support their conclusion that TRIC-A channel but not Tric-b is involved in cardiac rhythm formation and sympathetic nerve regulation of heart rate. Their data are of high quality. I have no any concerns or critiques.

>>>>>

Thank you for your comments. 

Reviewer #2: General comments:

Murakami et al. provide evidence for a crucial role of TRICA in cardiac pace making. Using murine genetic knock-out models, the authors demonstrate significantly reduced heart rate as a result ofTRIC-A ablation. In vivo measurements as well as experiments in isolated cardiac tissue including electrophysiological recordings in cells from the SA region demonstrate that physiologic pace making as well as adrenergic control of cardiac functions is strictly dependent on TRICA expression. The study is carefully planned and the open questions about a role of TRICA in physiologic control over heart rate are addressed by adequate and complementary methods. However, I have several issues regarding data presentation and analysis. With respect to the conclusion, I feel that the still enigmatic mechanism underlying the cardiac function of TRICA needs more detailed discussion, and some of the results, especially the measurements of PKA-mediated phosphorylation deserve more attention.

>>>>>>

We did not confirm a direct relationship between TRIC-A and PKA. Rather, we found decreased responsiveness to β-adrenergic stimulation in TRIC-A-deficient hearts. The manuscript has been revised according to the above comments.

Major points:

1. The authors show convincingly that in TRIC-A -/- mice not only heart rate is lower and barely increases in response to beta-adrenergic stimulation, as compared to WT or TTIC-B +/-, but also beta adrenergic inotropism is largely eliminated. This is an intriguing finding and rather unexpected. The original hypothesis of the authors is that TRICA is part of the cardiac ER/SR Ca2+ handling machinery and as such part of the coupled clock in pacemaker cells. Although this is still plausible and may not be ignored, adrenergic signaling is obviously impaired at the level of PKA-dependent phosphorylation as demonstrated in Figure 1A for CREB. Since PKA-mediated phosphorylation is an essential mechanism of adrenergic control over the coupled clock machinery (see Vinogradova and Lakatta 2009; DOI: 10.1016/j.yjmcc.2009.06.014) the question arises: Can the authors exclude that the main role of TRICA is to enable proper PKA-dependent regulatory phosphorylation in cardiac cells? It is in any case very important to further strengthen the finding that genetic ablation of TRICA impairs PKA regulatory phosphorylation. Please include a more informative/ representative immunoblotting illustration together with a decent statistics on CREB phosphorylation. I strongly suggest to further corroborate this important finding also for other targets (e..g. phospholamban), and include a more in depth discussion as to how TRICA might interfere with PKA signaling. In this respect one need to consider that TRIC has recently been identified as an interaction partner of STIM1 to organize STIM/Orai complexes (Shrestha et al 2020/ DOI: 10.1371/journal.pbio.3000700), which in turn are coupled to local cAMP, PKA/Ca2+ signaling (Zhang et al 2019 /doi.org/10.1038/s41467-019-09593-0). Thus the authors may discuss the role of TRICA in SA cells beyond the original countercurrent concept or direct modulatory effects on RyR signaling.

>>>>>>>>>>>

Intracellular calcium affects a number of channels, including HCN, CaV1.2, and NCX. The sinus node is complex and too small for detailed analysis. To determine the exact role of TRIC-A channels in PKA-dependent phosphorylation, it is necessary to perform analyses under simple conditions, such as TRIC-A overexpression in HEK293 cells. Nevertheless, this point is important.

We examined phosphorylation of phospholamban, which yielded results similar to those of the p-CREB analysis. Due to the low heart rate in TRIC-A−/− mice, physiological analysis of the response to adrenergic stimuli may yield erroneous results. Physiological evaluation is usually dependent on the hypothesis that basal intracellular conditions are almost the same between wild-type control and TRIC-A−/− mice. This may cause misinterpretation in TRIC-A−/− mice. 

Recently, Zhou et al., reported isoproterenol-induced arrhythmia in TRIC-A−/− mice. They demonstrated a marginal effect of isoproterenol on heart rate, which was consistent with the results of the present study. In addition, they reported that chronic treatment with isoproterenol resulted in cardiac fibrosis in TRIC-A−/− mice, which was reminiscent of chronic heart failure [Zhou et al., Circulation Research. 2020;126:417 – 435. Reference # 22 in the manuscript]. In their study, it seems likely that TRIC-A ablation resulted in exacerbated high sensitivity to β-adrenergic stimulation (arrhythmia and fibrosis), although basal heart rate was apparently lower than that of wild-type mice. If TRIC-A gene deletion causes high sympathetic responses under basal conditions, it may also cause a marginal response (i.e., increase in heart rate) to isoproterenol. Nevertheless, further studies are needed regarding the β-adrenergic cascade in TRIC-A−/− mice.

2. Along the same lines: STIM1 is a likely interaction partner of TRICA (Shrestha et al 2020). Moreover, STIM1/Orai signaling appears crucial (Zhang et al. 2015/ doi:10.1073/pnas.1503847112). Therefore, inspection of STIM1 and also Orai expression at the RNA level is recommended and the discussion should be extended towards a potential interference with the STIM/Orai pathway.

>>>>>>>>>

We additionally examined expression of STIM1, Orai1, Orai2, and phospholamban (S1 Fig). We previously analyzed STIM1+/− mice and found no significant changes in Heart Rate (HR) (Ohba et al., 2017, PLOS ONE | https://doi.org/10.1371/journal.pone.0187950).

>>>>>>>>>>>

3. The authors repetitively claim that „sympathetic nerve regulation“ is modified/impaired. From my understanding, there is no solid evidence for a role of TRICA in sympathetic neurons? By contrast, the authors demonstrate in Figure 4 almost complete elimination of isoproterenol-induced positive inotropism in isloated atria. These impressive effects can hardly be explained by sympathetic nerve regulation. The authors need to explicitly discuss the evidence for a neuronal vs a muscular/cardiac mechanism and omit all misleading statements.

>>>>>>>>>>>>>

TRIC-A−/− mice have an obviously decreased heart rate, which is independent of blood pressure, as isolated hearts also showed a decreased heart rate. We have revised the text accordingly.

Minor points:

1. The concentration of isproterenol (100 nM) should be stated also in the abstract.

>>>>>>>>>>>>>

The manuscript has been revised according to the reviewer’s comment.

>>>>>>>>>>>>>

2. Statistical analysis needs a more detailed description. What test(s) was/were used for comparison of multiple groups? The numbers of observations need to be stated consistently throughout the manuscript (eg. n is given with 9/10 in the text but stated as 6-15 in Figure 2A).

>>>>>>>>>>>>>

The manuscript has been revised according to the reviewer’s comment.

>>>>>>>>>>>>>

3. Specifically, statistics for immunoblotting data need extension. An N around 4 is in general insufficient for a decent statistical analysis. Statistical information, eg. as bar graphs, should be given in the respective figures.

>>>>>>>>>>>>>

We performed additional analyses and presented bar graphs in S1 Fig of the revised manuscript.

>>>>>>>>>>>>>

4. The designation of genetic models in text an figures is highly inconsistent – please use uniform terminology. Eg. In Figure 3, four(!) different terms are used for TRICA knock out mice: TricA-null, TRIC-A-null, TricA KO and TRIC-A- -/-.

>>>>>>>>>>>>>

The manuscript has been revised according to the reviewer’s comment.

>>>>>>>>>>>>>

5. In Figure 4, representative traces of contractile force are apparently of similar absolute amplitude and share the same scaling. However, the signal to noise ratio is substantially lower in the TRICA KO atria. Do the authors have any plausible explanation for this?

>>>>>>>>>>>>>

Unfortunately, force contraction was unstable, although we tried to avoid vibration and other factors to prevent noise. Nevertheless, better contraction traces are presented in the revised manuscript.

>>>>>>>>>>>>>

6. As above: Why are the TRIC-A -/- recordings in Figure 5 so noisy compared to WT?

>>>>>>>>>>>>>

Accurate action potential recording is key to this study. The trace was replaced with a better one in the revised manuscript.

>>>>>>>>>>>>>

7. The discussion may deserve remodeling – not only by including aspects mentioned above but also shortening other parts.

>>>>>>>>>>>>>

The manuscript has been revised according to the reviewer’s comment. 

>>>>>>>>>>>>>

8. Supplementary data are not reasonably mentioned in the text. The authors should clearly indicate, mention the results illustrated in the supplement.

>>>>>>>>>>>>>

The manuscript has been revised according to the reviewer’s comment.

---

## [Decision Letter · Decision Letter 1]

7 Dec 2020

Decreased cardiac pacemaking and attenuated β-adrenergic response in TRIC-A knockout mice

PONE-D-20-27381R1

Dear Dr. Murakami,

We’re pleased to inform you that your manuscript has been judged scientifically suitable for publication and will be formally accepted for publication once it meets all outstanding technical requirements.

Kind regards,

Agustín Guerrero-Hernandez

Academic Editor

PLOS ONE

Additional Editor Comments (optional):

Reviewers' comments:

Reviewer's Responses to Questions

**Comments to the Author**

1. If the authors have adequately addressed your comments raised in a previous round of review and you feel that this manuscript is now acceptable for publication, you may indicate that here to bypass the “Comments to the Author” section, enter your conflict of interest statement in the “Confidential to Editor” section, and submit your "Accept" recommendation.

Reviewer #1: All comments have been addressed

Reviewer #2: All comments have been addressed

2. Is the manuscript technically sound, and do the data support the conclusions?

Reviewer #1: Yes

Reviewer #2: Yes

3. Has the statistical analysis been performed appropriately and rigorously? 

Reviewer #1: Yes

Reviewer #2: Yes

4. Have the authors made all data underlying the findings in their manuscript fully available?

Reviewer #1: Yes

Reviewer #2: Yes

5. Is the manuscript presented in an intelligible fashion and written in standard English?

Reviewer #1: Yes

Reviewer #2: Yes

6. Review Comments to the Author

Reviewer #1: This is revised manuscript. The authors have addressed all concerns. No further comments. Congrats on such a nice work.

Reviewer #2: The authors did a good job in amending the manuscript. This is a highly valuable contribution to the field.

7. PLOS authors have the option to publish the peer review history of their article (what does this mean?). If published, this will include your full peer review and any attached files.

Reviewer #1: No

Reviewer #2: No

---

## [Editor Report · Acceptance letter]

10 Dec 2020

PONE-D-20-27381R1 

Decreased cardiac pacemaking and attenuated β-adrenergic response in TRIC-A knockout mice 

Dear Dr. Murakami:

I'm pleased to inform you that your manuscript has been deemed suitable for publication in PLOS ONE. Congratulations! Your manuscript is now with our production department. 

Kind regards, 

on behalf of

Dr. Agustín Guerrero-Hernandez 

Academic Editor

PLOS ONE